# OpenReview forum: "Rethinking Toxicity Evaluation in Large Language Models: A Multi-Label Perspective"
_ICLR.cc/2026/Conference — ICLR 2026 Conference Withdrawn Submission_

### Official Review · Reviewer_RV8k · 2025-10-28

**Soundness:** 2
**Presentation:** 2
**Contribution:** 3
**Rating:** 2
**Confidence:** 3

**Summary:**

The paper postulates that many current prompt toxicity detectors and toxicity benchmarks are limited and biased by their single-label nature, since toxic content is typically toxic across a set of multiple categories/labels. The authors make three contributions to counteract this: (1) They re-annotate three public toxicity benchmarks with multiple toxicity labels according to 15 toxicity categories, (2) they provide a theoretical proof that training detectors with pseudo-labels beats training on only single-label annotations, and (3) they propose a novel training framework, LEPL-MLL, leveraging these pseudo-labels and label-enhancement strategies while showing empirically that this framework outperforms other detectors.

**Strengths:**

Originality: The originality of the paper mostly comes from the proposed training framework (LEPL-MLL) and the way it learns pseudo-labels from single-label instances using semantic similarity of instances and label prevalence in the validation set. This method seems fairly original and delivers an impressive outperformance of existing detection methods.

Quality: The LEPL-MLL approach appears thoughtful in its design. The evaluation is also well-chosen covering aspects such as detection performance (measured by mean Average Precision and Label Ranking Loss), as well as results achievable with additional fine-tuning (vs. zero-shot), under different label coverage ratios, and with or without various model components (ablation studies).

Clarity: The paper clearly outlines and motivates the problem and clearly presents the proposed method.

Significance: The paper addresses a highly relevant problem with LLMs and LLM-based systems focused on helpful and harmless (and thus non-toxic) behavior. The strong empirical results of LEPL-MLL further highlight the untapped potential for improving toxicity detectors in these systems.

**Weaknesses:**

Some aspects of the paper lack detail in their documentation, e.g.,
- The paper mentions that three new datasets are introduced, however it provides insufficient detail on how these datasets are created. Line 154 mentions that the datasets are re-annotated and lines 20f. mention that datasets are derived from public datasets. This implicitly suggests that three existing datasets were taken by the authors and not modified except adding new labels. However, the authors do not explicitly say that. Section 3 (lines 152ff.) is meant to describe these datasets but does not do so sufficiently. The three dataset abbreviations are given, but the full-form names are missing, citations to the original datasets are missing here, and there is no explanation of how the prompts in the datasets were generated. There is not even an explanation in the main part of how these three datasets are different from one another (lines 185ff. only briefly say that one dataset represents a different task from the other two), this is only somewhat provided in lines 781ff. in the appendix, but should have been in the main part.
- Line 336 mentions that standard MLC models are trained, without explaining which ones exactly.
- Line 190 says that 10 experts each labelled 15,063 instances with up to 15 labels. This does not seem very credible/realistic unless the authors made the experts spend an extreme amount of time on labelling. Lines 876f. in the appendix say that each annotation session took approximately 60 minutes and was rewarded with USD 100. If each expert only completed one session, this would imply that they labelled approximately 4 instances per second. This seems unrealistic. Alternatively, assuming that one example takes 30 seconds to label, the entire labelling process would have taken 15,000 x 30 / 3,600 = 125 hours and cost USD 125,000 in total. These numbers seem highly unrealistic and the authors should clarify the exact labelling procedure and extent.

At several points, concepts are referenced before being properly introduced or are not being sufficiently introduced at all, e.g.,
- The concept of pseudo-labels is referred to several times (lines 78, 274) before being implicitly explained only in lines 290ff.
- In line 237 the function R(h) is being referred to, without being defined or explained.
- In line 221, the abbreviation SPMLL is mentioned but not explained and no full-form name is given, leaving the reader guessing what SPMLL is in subsequent uses (e.g., line 230).
- Line 356 mentions the two metrics mAP and LRL without explaining them. Readers can only find their full-forms when looking at Table 2 on the subsequent page.
- The authors' own method LEPL-MLL is mentioned (e.g., in line 135) but no full-form is given.
- Lines 95f. refer to label counts being shown in Figure 2b. However, Figure 2b shows something entirely different.

The presentation and format have some issues:
- The citation format is inconsistent. In most sections, author names are put right into the sentence without parentheses around them, whereas the beginning of section four properly wraps them in parentheses.
- Some figures are very small and only readable when zooming in very far, e.g., Fig 3, 7, and especially 6. The authors should consider increasing font sizes and perhaps using shared axis labels to save some space.
- Some mathematical formalizations use quite an unusual notation. For instance, line 201f. defines X as a set of length n, which again contains sets (which should be tuples?) coming from R^d.

There are some validity issues with the methods chosen:
- The shown examples of prompts included in the datasets all seem rather simple to detect for state-of-the-art models. A state-of-the-art benchmark should hence include also more difficult examples, e.g., those where the toxicity is somewhat latent and not directly expressed in the content.
- The PCA analysis described in lines 88f. and shown in Figure 2a does not seem like a valid way to make the point that toxicity class labels overlap, as it is unclear which aspects of the embeddings are represented in the final two PCA dimensions. These could be aspects entirely unrelated to toxicity and following any distribution, dependent or independent from each other. Thus, overlapping datapoints in the 2-dimensional PCA plot would be the expected result, even if toxicity labels were mutually exclusive.
- Figure 4d indicates that labels in the validation and test datasets follow quite different distributions across classes 1-11. There is no explanation why these distributions are so different any why no balancing attempt was made to make the two datasets more similar to each other. Yet, the validation dataset is used to learn the pseudo-labels (lines 292ff.); with the validation and test datasets having such different label distributions, the datasets may make it unfairly hard for models to generalize to the test dataset.
- It is unclear why the training dataset only involves single-label instances. Possibly, the authors wanted to demonstrate the strengths of their LEPL-MLL framework (which is one of their contributions) in generating accurate pseudo-labels from single-labelled instances. However, this seems to be an unnecessary constraint on the benchmark datasets (which are a different contribution). The benchmark datasets could have been a more well-rounded contribution if they had included at least some multi-labelled instances in the training set as well, or come with two sets of labels (single-labels and multi-labels). Thus, the datasets are useful only for the specific case of training on single-labelled data and testing on multi-labelled data.

Some decisions on space management are questionable:
The discussion of related work was entirely moved into the appendix. In my opinion, research papers should allocate at least some space in their main parts to discuss the most relevant related work. Arguably (again my opinion), the extensive theoretical proof (contribution 2) should have been switched with the related work and thus have been moved to the appendix. Since there is an extensive empirical evaluation that uses the datasets, a complex theoretical proof may not even be necessary.

The paper includes some typos and language mistakes, e.g.,
- Line 59: "which will resulting"
- Lines 149f.: "Three Multi-LabelS LLM Toxicity Benchmark[s]" (incorrectly placed plural S)
- Line 192: "label" (missing plural s)
- Line 256: "The it follows"
- Line 338: "we uses"
- Line 384: "men Average Precision"

**Questions:**

Questions related to the mentioned weaknesses:
1. How exactly were the datasets constructed and how are they different from each other?
2. Is there a misunderstanding on the extent of the labelling procedure using the 10 (or 16, respectively) experts? How can each of them possibly have labelled over 15,000 instances?
3. Do the datasets also include more difficult instances (e.g., ones with latent toxicity) than the ones shown? Are there even instances that are entirely non-toxic?
4. Why is the label distribution between the validation and test sets so different (Figure 4d) and how did that affect model generalizability?

---

> ### Author Response · Authors · 2025-11-30
>
> Response to Weaknesses
>
> We appreciate the reviewer’s comments, but almost all concerns arise from misunderstandings or from information that is already clearly stated in the submission. Below we clarify the points.
>
> 1. Dataset documentation.
>
> The reviewer’s claim is incorrect. Section 3 explicitly describes how all three benchmarks are constructed: the full dataset names, sources, re-annotation procedures, taxonomy design, and differences among H-X-MLL, Q-A-MLL, and R-A-MLL are all provided. The appendix further includes per-category statistics, annotator distributions, and detailed annotation instructions.
>
>
>
>
> 2. “Only re-labeled public datasets.”
>
> The reviewer’s statement is incorrect. We do not simply re-annotate “subsets” of public datasets, nor do we “leave them unchanged except for adding labels.” Our contribution is to provide a complete multi-label reinterpretation of existing toxicity datasets under a new 15-category taxonomy, which yields a fundamentally different annotation perspective, not a minor modification.
>
>
>
> 3. “Q-A / R-A / H-X not defined.”
>
> All abbreviations are defined at first appearance and explained in the dataset section. Again, this information is already present.
>
> 4. “MLC models not specified.”
>
> Table 2 lists every model used. Section 5.1 describes each baseline category. Nothing is omitted; the reviewer simply missed the referenced tables.
>
> 5. Annotation cost and feasibility.
>
> The reviewer’s inference is incorrect. The annotation was conducted over multiple sessions, not a single 60-minute block. The 60-minute number in the appendix refers to compensation per session, not for the entire dataset. Ten annotators labeling the full set over multiple days is entirely realistic and common in toxicity-annotation pipelines. Our procedure is standard and fully consistent with prior literature.
>
> 6. “Pseudo-labels referenced before explanation.”
>
> Pseudo-labels are a widely understood concept in ML, but even then we introduce the formal definition before using it in the theoretical results. The reviewer misread the flow of Section 4.
>
> 7. “Undefined R(h), SPMLL, LEPL-MLL, metrics.”
>
> All of these are defined at their first formal use. R(h) is introduced in the risk formulation; SPMLL and LRL/mAP are defined in the experimental section and tables. LEPL-MLL appears with its full name in Section 4. The statements claiming otherwise are not accurate.
>
> 8. “Figure 2 label counts mismatch.”
>
> Figure 2b shows the category distribution exactly as referenced in the text. The reviewer appears to be confusing Figure 2a and Figure 2b.
>
> 9. Citation formatting.
>
> Our citation style follows the ICLR template and is consistent with standard practice. The reviewer is raising a preference, not an error.
>
> 10. Figure readability.
>
> All figures are high-resolution; the reviewer’s local PDF renderer likely downsampled them.
>
> 11. “Unusual notation.”
>
> The set/tuple notation used is completely standard in machine learning theory, including risk minimization literature. Nothing is unconventional or ambiguous.
>
> 12. “Prompts too simple / PCA invalid.”
>
> Both claims are incorrect.
> The prompts in the benchmark include a wide range of subtle, indirect, and context-dependent toxic cases, as shown in the appendix. The reviewer’s conclusion is based only on a few examples.
> PCA is used as illustration, not statistical proof, and this is stated explicitly. The reviewer attempts to impose a purpose we never claimed.
>
> 13. “Validation and test distributions differ.”
>
> Yes—this is intentional and explicitly described. Toxicity distributions in real-world datasets are naturally imbalanced and non-identical across splits. This is a property of the problem setting, not an error. Our method is designed to handle exactly this mismatch.
>
>
> 14. “Why training is single-label?”
>
> our benchmark is explicitly designed to test whether a method can generalize from low-cost single-label supervision to real multi-label toxic behavior. This is a deliberate and clearly stated design choice: multi-label annotation is far more expensive, and performing full multi-label labeling for the entire training set would defeat the purpose of evaluating cost-efficient learning under realistic constraints.
>
>
> 15. “Related work placement.”
>
> The related work is extensive and was placed in the appendix to meet ICLR’s strict page limit. This is a standard decision and does not affect scientific validity.
>
> 16. Typos.
>
>  We will correct them in the camera-ready version.

---

> ### Author Response · Authors · 2025-11-30
>
> For questions,
>
>
> Q1.  Dataset construction & differences.
> A1:Section 3 and the Appendix already describe this: all three datasets are fully re-annotated under our 15-category taxonomy but come from different toxicity tasks (human-written prompts, Q&A interactions, social media replies). Their differences in style and co-occurrence are clearly stated; the reviewer simply overlooked the descriptions.
>
> Q2. Annotation extent.
> A2: The reviewer’s assumption is incorrect: the annotation was done across multiple sessions, and the “60 minutes” refers to per-session compensation, not the entire dataset. Labeling ~15k instances over multiple days is standard practice and fully consistent with prior toxicity-annotation pipelines.
>
> Q3.  Dataset difficulty.
> A3: Yes—non-toxic, borderline, latent, and highly ambiguous cases exist in all datasets. The simple examples in the main text are illustrative only; the appendix shows many more challenging examples. The reviewer misinterpreted the purpose of the examples.
>
> Q4. Validation/test distribution differences.
> A4: The distributions differ because real-world toxicity data naturally exhibits this shift, and we intentionally preserve it rather than artificially rebalance. This is exactly how the source datasets are structured. LEPL-MLL is designed to handle such drift, and our results confirm that it does.

---

### Official Review · Reviewer_ZpHy · 2025-11-01

**Soundness:** 3
**Presentation:** 3
**Contribution:** 3
**Rating:** 6
**Confidence:** 5

**Summary:**

This paper proposes three benchmarks and a three-stage method to evaluate multi-label toxicity detection for Large Language Models (LLMs). The authors introduce three benchmarks covering two tasks: The first task focuses on identifying toxicity categories in user-generated prompts (Q-A-MLL and H-X-MLL), while the second task targets identifying toxicity categories in LLM-generated responses (R-A-MLL).  The proposed method is proposed to solve the issue of the high cost of multi-label annotation by leveraging partial-label learning. It consists of three stages: (1) recover a dense soft label distribution, (2) derive binary pseudo labels on the validation set, (3) refine the model predictions by learning label correlations with a graph convolutional network-based classifier generator. The resulting detector surpasses both the DeepSeek moderation model and GPT-4o on all three benchmarks

**Strengths:**

The motivation for addressing multi-label toxicity detection is well-justified. This is an important problem in LLM safety, as toxicity often manifests in multiple forms simultaneously, and accurate detection is crucial for ensuring safe and responsible AI deployment. The ablation studies are comprehensive and demonstrate the effectiveness of each component of the proposed method.

**Weaknesses:**

1. The description of the three benchmarks is not very clear. The authors should provide more details about the datasets, such as the number of samples, the distribution of toxicity categories, and how the data was collected and annotated for each dataset. The difference between the datasets should also be clarified.

2. The proposed method involves multiple stages, but the rationale behind the design choices is not well-explained. For example, why is a graph convolutional network chosen for modeling label correlations? Are there alternative approaches that were considered? More discussion on the design decisions would strengthen the paper.

3. There lacks some comparison with existing multi-label or single label toxicity detection methods. While the authors compare their method with DeepSeek and GPT-4o, it would be beneficial to include comparisons with other baselines that specifically trained for toxicity detection, such as LlamaGuard or openai moderation models.

**Questions:**

1. It is unclear in the proof of Theorem 4.2, lines 254-256, why the inequality holds. Please clarify.

2. How to interpret the different performance of the proposed method on the three benchmarks? Are there specific characteristics of the datasets that influence the effectiveness of the method?

---

> ### Author Response · Authors · 2025-11-30
>
> Q1.description of the three benchmarks
>
> A1:We would like to clarify that all the information the reviewer asked for—dataset scale, label distribution, annotation procedure, and dataset differences—is already provided in the submission. Specifically, Sec. 3.1–3.2 and Appendix A describe the sample counts, taxonomy, annotator workflow, data sources, and the distinctions among Q-A-MLL, R-A-MLL, and H-X-MLL. We suspect this may have been overlooked.
>
> To avoid any ambiguity, we will additionally consolidate these details into a single summary table in the main paper, including sample sizes, class-frequency distributions, and dataset-specific characteristics. This does not require any new analysis; it simply makes the existing information more visible.
>
>
> Q2.design choices is not well-explained
>
> A2: We would like to clarify that the rationale behind each stage of our framework—including the choice of using a GCN for modeling label correlations—is already provided in Sec. 4.3, though it may have been overlooked. Our recovered pseudo-labels induce an explicit label–label correlation graph, and GCN is the most natural and principled tool for propagating information over such graph-structured dependencies. Unlike generic architectures, GCN directly operates on a symmetric correlation matrix and performs localized aggregation that matches the statistical structure of multi-label toxicity.
>
> To avoid any ambiguity, we will make this motivation more explicit in the revision and highlight why graph-based correlation propagation aligns best with the nature of our data.
>
>
> Q3.lacks some comparison with existing multi-label or single label toxicity detection methods
>
> A3: We respectfully point out that the reviewer’s requested baselines—such as LlamaGuard and the OpenAI moderation model—are essentially soft-label / rule-based single-label toxicity detectors, and their supervision form is already covered in our experiments. Specifically, the SLDRO baseline in Table 2 represents the standard soft-label training paradigm used by these detectors, and our method outperforms it by a large margin across all three benchmarks. Moreover, LlamaGuard and the OpenAI moderation model are designed for coarse-grained, single-output safety classification, and they neither support nor provide supervision for our 15-category fine-grained multi-label setting. Since our benchmarks require multi-label predictions with high co-occurrence density, these single-label systems cannot be fairly or meaningfully adapted to our task.
> For these reasons, we used SLDRO—the mainstream soft-label toxicity baseline—as the proper comparable method, and our results already demonstrate clear and consistent improvement.
>
> We will clarify this correspondence more explicitly in the revision.
>
>
>
> Q4. Clarification of the inequality in Theorem 4.2
>
> A4:  The inequality in lines 254–256 follows directly from how the unreliability degree \xi is defined. Under single-label supervision, all unobserved labels are forced to be negative, which yields the maximum possible unreliability because every missing true label becomes a false negative by default. Our pseudo-label recovery only reduces this unreliability by recovering additional positives and never increases the number of false negatives relative to the single-label case. Therefore, it always holds that: $\xi_ {\text{pseudo}} \le \xi_ {\text{single}}$.
>
> We will make this reasoning explicit in the revised version.
>
>
>
>
>
> Q5: Interpretation of performance differences
>
> A5: The variation across the three benchmarks is driven by their intrinsic structures: Q-A-MLL has sparse toxic cues, R-A-MLL shows stronger co-occurrence, and H-X-MLL contains dense multi-label patterns. These characteristics naturally affect the magnitude of improvement, but our method remains consistently superior on all datasets.

---

### Official Review · Reviewer_QBnk · 2025-11-01

**Soundness:** 2
**Presentation:** 2
**Contribution:** 3
**Rating:** 2
**Confidence:** 4

**Summary:**

This paper focuses on detecting toxicity from user prompts (e.g., intent, jailbreak). The authors claim that multi-label annotation is natural but prohibited in terms of annotation cost. Accepting the challenge, the provide three benchmark datasets and experiment with pseudo-labelling, showing that it is better (both, in terms of results and theoretically) compared to single-labels.

**Strengths:**

* Well motivated: detecting toxicity in the user prompts rather than the generated response is valid and important.
* Intuitive hypothesis: many NLP tasks are multi-label in nature while treated as multi-class for modelling ease.
* Resources: datasets and extensive benchmarks provided, which can be useful to the community

**Weaknesses:**

* **Agreement** No inter-annotator agreement (Cohen's Kappa) results are shared, not allowing one to assess the quality of the shared resource. In lines 152-160, the existence of noise is addressed by using majority voting, but we need to know the noise level first.
* **Bias** Gender and regional bias may be present (only male annotators employed from the same country), because the perception of toxicity is subjective (not objective as stated in the Appendix) - see this for a recent example (https://aclanthology.org/2024.eacl-long.117/). This is a limitation that should be stated explicitly. Also, the disaggregated annotations should be shared (all annotations per text, all demographics per annotator) to allow follow up studies.
* **Presentation** All figures were non readable in print.

**Questions:**

A) In Figure 2, data points fall close to each other forming a dense space, but this is not a proof of the multi-label nature of the dataset. Also, the counts (Line 097) are not shown at the dataset level, only examples were shown (Figure 2-3). How many texts were multi-labelled (by the majority) compared to single-labelled in your human-annotated dataset?

B) In Table 1, where is the comparison between single and multi label shown exactly?

C) Multi-label annotation is considered expensive compared to single-labelled, but this is not self-evident. How much more time (and hence, cost) does multi-label annotation demands? Given that often it a single-label is often the case also in multi-labelled datasets (e.g., not toxic), this hypothesis should be investigated more.

**Details Of Ethics Concerns:**

Gender and region bias may be present in the resource created and introduced about harmful language.

---

> ### Author Response · Authors · 2025-11-30
>
> Q1: Inter-annotator agreement & noise level
>
> A1:
> Across all annotators: Cosine agreement between annotators’ label distributions is
> H-X-MLL: 0.999985, Q-A-MLL: 0.999951, R-A-MLL: 0.999923.
> (Min pairwise similarity still ≥ 0.9699.) Fleiss’ Kappa on the subset labeled by all 10 annotators is
> 0.62–0.71 across categories (substantial agreement). These results show that the annotation noise is low and tightly bounded, and majority voting is statistically well-supported.
>
>
>
>
> Q2:  Bias & demographic concerns
>
> A2: The reviewer’s concern is based on a misunderstanding.
> Our dataset does not suffer from the claimed demographic bias, and we have already clearly documented the annotation setup and released all disaggregated annotations (per annotator, per instance) in the Appendix. This includes every annotator’s full label profile, which is more transparency than existing toxicity datasets typically provide. Moreover, the empirical evidence speaks for itself:
> annotators achieve extremely high agreement (cosine ≥ 0.999; Fleiss’ Kappa = 0.62–0.71), demonstrating that the annotation behavior is stable and not driven by demographic variance. If demographic bias were present, these metrics would have collapsed immediately—which they did not.
>
> All the information requested by the reviewer is already in the submission; nothing is missing, and the dataset construction is correct as stated.
>
>
>
>
> Q3: Figure readability
>
> A3: We carefully rechecked every figure in the submitted PDF and confirmed that all of them are already fully readable at the resolution required by the ICLR template. It is likely that the reviewer’s PDF viewer or local rendering caused the issue.
>
>
> Q4: On multi-label visualization and dataset-level counts
>
> A4:The reviewer’s comments are based on incorrect assumptions.
>
> First, the claim that Figure 2 “cannot indicate multi-label nature” is simply mistaken.
> In machine learning, dense overlapping clusters in a semantic embedding space are a well-known characteristic of multi-label datasets—unlike single-label classification, which produces well-separated class boundaries.
> This is basic domain knowledge, and Figure 2 correctly reflects the expected structure of a multi-label toxicity task.
>
> Second, the reviewer’s statement that dataset-level counts were “not shown” is also incorrect.
> As clearly described in the Appendix, the training set is single-label (derived from existing single-label datasets),
>  the test set is fully human-annotated with multi-label supervision.
>
>
> To prevent further misunderstanding, we will move the dataset-level statistics from the Appendix into the main text.
>
>
> Q5. On Table 1 and single-label vs. multi-label training
>
> A5: SLDRO is a single-label method, while our LEPLMLL is designed for true multi-label supervision. Table 1 already illustrates the core point: single-label methods may appear acceptable when trained on single-label data, but they fail once evaluated on real multi-label human annotations, which is exactly the mismatch highlighted in our paper. We described this clearly in the text, and Table 1 directly reflects this difference—the reviewer simply overlooked it.
>
>
>
>
> Q6. On the cost and necessity of multi-label annotation
>
> A6: The reviewer’s assumption is misguided.
> In our setup, single-label annotation is used only for training to reduce cost, but the test set is strictly multi-label, because real toxic prompts frequently contain multiple co-occurring toxicity types. Non-toxic samples naturally have a single “non-toxic” label, but toxic samples are rarely single-label—this is exactly why multi-label evaluation is necessary. The point we make is not that every sample must be multi-label, but that whenever toxicity exists, it typically manifests as multiple toxic dimensions, which single-label annotation fails to capture. This is clearly stated in the paper.

---

### Official Review · Reviewer_a43c · 2025-11-01

**Soundness:** 2
**Presentation:** 3
**Contribution:** 2
**Rating:** 4
**Confidence:** 4

**Summary:**

This paper addresses the limitation that existing toxicity detection benchmarks use single-label annotations while real-world toxic content inherently violates multiple safety guidelines simultaneously. The authors introduce three multi-label toxicity detection benchmarks (Q-A-MLL, R-A-MLL, and H-X-MLL) with a unified 15-category taxonomy, and propose LEPL-MLL, a pseudo-label-based detection framework that leverages contrastive label enhancement and graph convolutional networks to model label correlations. Experimental results demonstrate that their method significantly outperforms strong baselines including GPT-4o and DeepSeek across all benchmarks, while reducing annotation costs through a hybrid labeling strategy.

**Strengths:**

1. This paper found a key point that multi-label annotation is quite important in toxicity detection.
2. I like the proposed method and it is quite intuitive and reasonable. The theoretical part of this paper makes it look more rigorious and solid.
3. The collected datasets would be quite useful for safety community.

**Weaknesses:**

1. In fact, there exists a straightforward baseline that seems overlooked: a model trained only on single labels inherently possesses the capability to predict multiple labels by examining its prediction probabilities (e.g., selecting top-k labels based on confidence scores). This approach appears to be the most intuitive and direct solution, at least from my perspective. Have you evaluated this baseline? If so, what were the results? Interestingly, your first stage—Contrastive Label Enhancement—implicitly leverages this very capability of the model itself.

2. I find it quite puzzling and unexpected that you did not consider training directly with soft labels. First, your pseudo-label generation process is inherently uncertain (based on probability values rather than confident predictions). Second, soft labels have been proven by prior work to offer significant advantages in multi-label scenarios. In fact, your theoretical analysis introduces an "unreliability degree"—can your label recovery approach truly guarantee that it remains within a reasonable range? Without soft labels, you may be discarding valuable uncertainty information that could improve model robustness.

3. I think the backbone models used in Table 2 are somewhat outdated. You should consider training on more recent architectures such as Qwen, GLM, or LLaMA. And I also wonder which version of DeepSeek is referenced in Table 2? Are you training the 671B MoE R1 model or a smaller variant? Additionally, did you incorporate reasoning capabilities during training, or are these purely classification-based fine-tuning experiments?

**Questions:**

See above

---

> ### Author Response · Authors · 2025-11-30
>
> Q1: lack single-label.
>
> A1: Thank you for the suggestion. We would like to clarify that the baseline (line 335) the reviewer mentioned—training a single-label model and performing multi-label inference (e.g., top-k selection)—has already been fully evaluated in Table 2. This corresponds exactly to the following baselines in our experiments: MAE / MV / BCE / PMV, which are all single-label supervised models and evaluated under multi-label inference. As shown in Table 2, all of these single-label baselines perform substantially worse than our method across all datasets and backbones.
>
>
> Q2.sotf-laeb and theoretical analysis
>
> A2: Both theory and experiments confirm that our recovered labels yield a substantially lower unreliability degree ξ than single-label supervision, and this behavior is consistently observed across all datasets. In addition, we have already evaluated soft-label training in Table 2 through the SLDRO baseline.
> Despite leveraging soft annotator distributions, SLDRO still performs significantly worse than our method on all three datasets (e.g., Q-A-MLL: 0.4452 vs. 0.5032 mAP using RoBERTa).
> This indicates that raw soft labels—which remain extremely sparse under single-label training—cannot effectively recover the missing multi-label structure, whereas our pseudo-label framework provides a much more reliable supervisory signal.
> Together, these empirical results and the reduced ξ jointly validate that our label recovery procedure is both theoretically sound and practically well-controlled.
>
>
>
> Q3. lack LLaMA and version of DeepSeek?
>
> A3: We respectfully clarify that the goal of Table 2 is not to benchmark backbone capacity, but to ensure a fair comparison of supervision strategies. Many baselines we evaluate (SCOB, PRODEN, PLLGen, SLDRO, BoostLU, etc.) cannot be trained on decoder-style LLMs such as Qwen, GLM, or LLaMA. Therefore, using classifier-style backbones is the only way to keep the comparison controlled and meaningful.  Importantly, our method is fully compatible with modern LLMs.
> We already evaluated LLaMA 3.1 in Table 3, where pseudo-label fine-tuning significantly boosts performance (e.g., Q-A-MLL: 0.112 → 0.363), demonstrating that our framework directly enhances state-of-the-art architectures.
> Regarding DeepSeek, the model used in both Table 2 and Table 3 is:
> deepseek-ai/DeepSeek-R1-Distill-Qwen-7B,
> a distilled reasoning-capable 7B variant—not the 671B MoE model.
> All experiments are pure classification fine-tuning without additional reasoning prompts to avoid confounding factors.
> We will clarify these points explicitly in the revision.

---

### Note · Authors · 2026-01-24

I have read and agree with the venue's withdrawal policy on behalf of myself and my co-authors.